# Stakeholders perspective of integrating female genital schistosomiasis into HIV care: A qualitative study in Ghana

Emmanuel Asampong[1], Franklin N. Glozah[1], Adanna Nwameme[1], Ruby Hornuvo[1], Edward Mberu Kamau[2], Philip Teg-Nefaah Tabong[1]*

1 Department of Social and Behavioural Sciences, School of Public Health, College of Health Sciences, University of Ghana, Legon, Accra, Ghana, 2 UNICEF/UNDP/World Bank/WHO Special Programme for Research and Training in Tropical Diseases (TDR) at World Health Organisation, Geneva, Switzerland

* ptabong@ug.edu.gh

## Abstract

### Introduction

In Sub-Saharan Africa (SSA), HIV remains the leading cause of adult premature death. The rising prevalence of Female Genital Schistosomiasis (FGS) in SSA, including Ghana, has led to a growing dual burden of HIV-FGS cases. This trend has prompted the WHO to advocate for integrated HIV and FGS services. This study examined stakeholder perspectives on integrating FGS prevention and control with HIV care in endemic areas of Ghana.

### Methods

The study took place in Ga South Municipality, Greater Accra Region, Ghana. A qualitative approach combining narrative and phenomenological designs was used. Data collection included Focus Group Discussions with Community Health Officers (CHOs) (n = 9), and Key Informant Interviews with healthcare providers at regional, district, and community levels (n = 13). In-depth interviews were also conducted with individuals affected by FGS and HIV (n = 13), female household members (n = 10), Community Health Management Committee members, and community leaders (n = 7). Participants were purposively selected. Audio-recorded interviews were transcribed, coded, and thematically analyzed using NVivo version 13.

### Results

There was a notable knowledge gap on FGS among CHOs and community members. Many health workers mistook FGS for sexually transmitted infections, while community members primarily recognized it through gynecological symptoms. Healthcare was sought from a mix of formal health facilities, herbalists, and spiritual

**Data availability statement:** All data are in the manuscript and/or Supporting information files.

**Funding:** This work was supported by the Access and Delivery Partnership (ADP) from the Government of Japan, the United Nations Development Programme, World Health Organization, the Special Programme for Research and Training in Tropical Diseases (TDR) and PATH research grants to EA, FNG, AN, PT-NT (grant ID: 2023/1408508-0). The funders had no role in study design, data collection and analysis, decision to publish, or preparation of the manuscript.

**Competing interests:** The authors declare that no conflict of interest exists.

centers, often delaying accurate diagnosis and management. Barriers to integrating HIV and FGS services included limited awareness, stigma, cultural beliefs, provider attitudes, and resource shortages.

## Conclusions

Both CHOs and community members lacked sufficient knowledge about FGS, hindering regular screening and timely diagnosis. While integrating FGS and HIV care could support Ghana's HIV eradication goals, success depends on addressing stigma, improving awareness, ensuring drug availability, and equipping health facilities. Collaboration among healthcare professionals and developing standardized clinical protocols are essential. Training community health workers on these protocols is urgently needed to support effective integration.

## Author summary

In Sub-Saharan Africa (SSA), HIV remains a leading cause of premature adult death. Recent data suggest a link between rising HIV prevalence and the global increase in Female Genital Schistosomiasis (FGS), particularly in SSA countries like Ghana. The emergence of co-existing HIV-FGS cases has prompted the World Health Organization (WHO) to call for integrated services. This study explored stakeholder perspectives on integrating FGS prevention and control with HIV care in endemic areas of Ghana, specifically the Ga South Municipality in the Greater Accra Region. Qualitative interviews and discussion were conducted with Community Health Officers, healthcare stakeholders, individuals living with FGS, female household members, and community leaders. Participants were purposively selected, and data were thematically analyzed using NVivo 13. Findings revealed limited knowledge of FGS among both community members and health workers, with frequent misclassification of FGS as a sexually transmitted infection. Barriers to integration included stigma, resource shortages, provider attitudes, and cultural beliefs. The study recommends developing clinical protocols, training health workers, and enhancing collaboration to support early detection and integrated care for FGS and HIV.

## Introduction

Schistosomiasis, which is also known as "snail fever" or bilharzia [1], is a water-based debilitating Neglected Tropical Disease (NTD) of poverty, that affects an estimated 250 million people globally with 97% of all infections and 85% of the global at-risk population concentrated in sub-Saharan Africa [2]. In this region, almost 56 million girls and women are estimated to be affected by Female Genital Schistosomiasis (FGS) [3]. The disease, caused by trematode parasites of the genus *Schistosoma,* and three main species – *Schistosoma japonicum, S. mansoni* and *S. haematobium*

commonly infect humans [4]. Human schistosomiasis is transmitted in areas where fresh water is contaminated with schistosome larvae. Two major forms of schistosomiasis are intestinal and urogenital. *Schistosoma haematobium* is the species most associated with urogenital schistosomiasis, whereas intestinal schistosomiasis may also be caused by five other species: *S. guineensis, S. intercalatum, S. japonicum, S. mansoni and S. mekongi*. Approximately two thirds of all cases of schistosomiasis are attributed to infection with S. haematobium. These eggs in the urinary and reproductive organs cause inflammation and lesions [1].

The World Health Organization (WHO) in 2020 published a roadmap to help progress the control and elimination of NTD by 2030 [5]. The WHO in their recommendation mentioned the elimination of schistosomiasis as a public health problem and ending the transmission of schistosomes in human hosts in selected countries by 2030 by considering the WASH strategy (providing safe water, sanitation, and hygiene), the control of snail populations, and preventive chemotherapy. Preventive chemotherapy with praziquantel remains an effective approach for combatting the transmission of schistosomes in humans and must be administered annually in endemic settings [6].

Ghana carries one of the highest burdens of schistosomiasis in sub-Saharan Africa [7]. An estimated 3.1 million people are at risk of schistosome infection in endemic areas in Ghana [8]. In Ghana, reports indicate an estimated country-wide prevalence of 23.3%, with localized prevalence levels of more than 50% with over 400 communities in 18 districts from eight regions living in the Volta Basin affected by schistosomiasis while over 3.1 million are at risk of infection in the endemic areas alone in the Volta [8]. The Schistosoma haematobium, which causes urinary (FGS) schistosomiasis, is the predominant human schistosome species in Ghana, and is widely distributed in the country. In communities where schistosomiasis is endemic, it is estimated that between 33–75% of women have FGS yet few of these cases are diagnosed correctly [9]. Studies show that women with FGS may have a 3–4 times greater risk of contracting HIV [10,11]. In addition, epidemiological studies show that FGS is responsible for up to a three- to four-fold increase in horizontal transmission of HIV and AIDS [11–13], whereas a regression analysis of prevalence of S. haematobium infection and HIV in sub-Saharan African countries found that each S. haematobium infection per 100 individuals resulted in a 3% relative increase in HIV prevalence [14]. Given the high prevalence and incidence of FGS and its strong geographic overlap with HIV/AIDS in countries such as Ghana, and elsewhere, it stands to reason that FGS would be identified as a leading HIV/AIDS cofactor in Africa, and that mass drug administration (MDA) with the antiparasitic drug, praziquantel, would represent an important strategy for HIV/AIDS prevention.

Currently, services for HIV and FGS are provided separately thereby leading to service duplication. However, given the substantial resources deployed and health systems developed during the roll out of the global antiretroviral treatment (ART) [15], benefits from integrating HIV and FGS care would be two-fold. First, it would better address HIV and FGS prevention, management, and control; second, it would strengthen the health system and multisectoral approach, which is in line with Ghana's Neglected Tropical Disease (NTD) programme mandate to progressively reduce morbidity, disability and mortality using integrated and cost-effective approaches with the view to eliminate NTDs in Ghana. Overall, a community-based integration approach would improve access to basic diagnostics and essential medicine and referral systems, to ensure equitable care for FGS, and the risk of HIV. Therefore, to influence schistosomiasis control policy, through evidence-based studies, to direct efforts at women in reproductive age in Ghana, this study will seek to explore stakeholders' perspectives on the integration of prevention and control measures for FGS into HIV care, thereby contributing to the realization of the Sustainable Development Goals (SDGs), particularly good health and well-being (SDG3), gender equality (SDG5) and reducing inequality (SDG10).

## Materials and methods

### Ethics statement

The protocol for the study was reviewed and approved by the Ghana Health Service Ethics Review Committee (GHS-ERC: 001/01/24). All four research assistants who assisted in the data collection were trained in accordance with the study protocol prior to commencement of data collection.

Written informed consent was provided by all study participants prior to data collection. All study participants were assured of confidentiality and that the information provided will be reported with complete anonymity.

## Study setting

The study was conducted in the ten communities in the four sub-districts (Amanfro Sub Municipal, Bortianor Sub Municipal, Kokrobite Sub Municipal and Obum Sub Municipal) of the Ga South Municipality. The study communities include Manheam Nsuonanu, Kingstown, Amanfro, Tomefa, Brigade, Azumah Farms, Adakope, Machighani, Omankope and Gallilea Market.

The Ga South Municipality is one of the municipalities, located in the Southwestern part of the Greater Accra Region with its capital being Ngleshie Amanfro. The Municipality shares boundaries with Weija-Gbawe Municipality to the East, Upper West Akim to the North, Gomoa East to the South-West, Awutu-Senya East to the West, Awutu-Senya West to the North West and the Gulf of Guinea to the South [16]. According to the 2021 Population and Housing Census, the Ga South Municipality has a total population of 350,121, with 172,492 males and 177,629 Females [17].

There are two main rivers within the Municipality, the *Densu* and *Ponpon* rivers. The larger of the two is the Densu which drains down from the Eastern region through the western portion of the Municipality where it enters the sea. These water bodies are the major source of water for people in this community. Most community members especially women use the rivers for bathing, washing of clothes and cooking. The Densu River serves as a major economic source for women who usually sell fresh fish from the Densu River [16].

The Ga South Municipality was chosen for this study due to the high prevalence rate of FGS among the population. FGS is endemic within the Ga South Municipality especially within communities along the river [18–20] and it is well known among community households who usually refer to it as "the water-borne disease" (Fig 1).

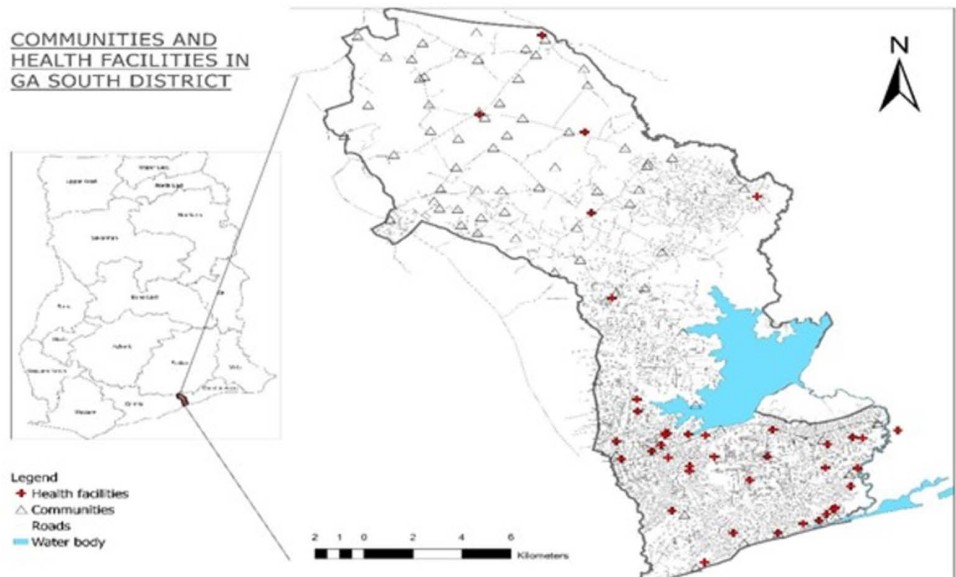

**Fig 1. Map showing study area with water bodies.** Source: Created with ArcGIS (basemap shapefile: Ghana statistical service: https://data.humdata.org/dataset/cod-ab-gha).

## Study design

The study adopted phenomenology and narrative approach of qualitative research to explore stakeholders' perspective on integrating Female Genital Schistosomiasis care into HIV/AIDS care in Ghana. The study adapted the Anderson's Behavioral Health Model of Health Service Utilization (Healthcare Utilization Model). This framework has been proven to be a useful model for studying and understanding factors that influence the use of various types of health services [21]. The model consists of three main components: the predisposing factors, enabling factors, and need factors.

Predisposing factors which are factors such knowledge about FGS and HIV and AIDS or attitudes towards health care providers and migration from one place to another can affect the use health services or influence an individual/community's willingness to utilize the health services. In this case, individual or community predisposing factors can affect the health system factors under the enabling factors in a way that, if individual attitudes towards tuberculosis contact tracing is negative, it will negatively affect the health system in a way that, the attitudes of the health care workers may be influence negatively, which might discourage them from conducting contact tracing. Predisposing factors can also directly influence integration.

Enabling factors such as resources availability, number of health care workers available, healthcare workers attitudes, practices/awareness can facilitate or enable one to access health care services. In this case, the enabling factors directly affect the predisposing and the health need factors. For example, if there are no resources for integration and other related essential services, someone may choose to migrate to areas where there are adequate resources. Or if there are inadequate resources for integration, then the health status or condition may also be affected.

Need factors such as perception, health status and quality of life can determine an individual's perceived need for health care services. These factors directly influence integration. For example, if there is a presence of FGS signs, or there is a change in the quality of life of an individual due to condition, they may choose to utilize the health service. These constructs informed the design of the data collection tools.

## Study population

The study population consisted of household heads, community members, community leaders, community health workers, and patients within the four sub-districts of the Study district in the Greater Accra Region of Ghana.

## Inclusion and exclusion criteria

Any person 18 years and above, who has lived in the community for more than one year is eligible to take part in the study. For community members, that individual should have lived in the community for more than one year. Health care workers who were actively involved in the provision, coordination, or management of healthcare services related to HIV and/or Female Genital Schistosomiasis (FGS) in Ghana were included in the study. To qualify, these individuals were required to have at least one year of experience in their respective roles within the health system. This included professionals working at the community, district, regional, or national levels, such as nurses, medical officers, public health officers, and health program coordinators, who directly contributed to service delivery, supervision, or policy implementation concerning HIV and/or FGS.

For participants living with HIV and/or FGS, eligibility was limited to individuals who had received a clinical diagnosis of either or both conditions and were currently enrolled in treatment. Specifically, individuals with HIV had to be on antiretroviral therapy (ART) at the time of data collection, and those with FGS had to have received or be receiving medical care for schistosomiasis-related symptoms. This ensured that participants could provide informed perspectives based on their lived experiences with ongoing care and treatment. Those who did not consent to participating were excluded from the study. In addition, participants who were very ill at the time of data collection were excluded from the study.

## Sample size and sampling technique

The sample size for this study conducted among key stakeholders was n = 52. This included a Focus Group Discussion (FGD) with Community Health Officers (n = 9), Key Informant Interviews (KIIs) with key stakeholders within the health sector of Ghana (n = 13) and In-depth Interviews with key stakeholders (n = 30) within the study communities which included Persons with FGS and HIV, Female Households, Community leaders and Community Health Management Committee members.

The study participants were purposively sampled (stratified purposive) to achieve the objectives of this study considering their key role as stakeholders in provision and/use of care for FGS and HIV/AIDs at the individual, community and regional/national levels and within the health sector as well as their roles in the reduction of disease burden of FGS and HIV/AIDS in Ghana.

## Data collection tools and strategy

**Focus group discussions (FGDs).** To contextualize the integration of FGS and HIV care into HIV continuum of care in community and health facility settings, an FGD was conducted among Community Health Officers (n = 9) from all the 10 communities in the Municipalities using an FGD guide to explore care provision for FGS and HIV, community involvement, and the barriers and enablers of integrating FGS and HIV into PHC.

**Key informant interviews (KIIs).** Using a KII guide, we conducted KIIs (n = 13) with various stakeholders to explore the feasibility, challenges, and opportunities of integrating FGS prevention and control package with HIV continuum of care in communities. Also, challenges and barriers to the delivery of FGS prevention and control package with HIV. These include Health care professionals at national, regional, district and community levels, including CHPS.

**In-depth interviews (IDI).** In-depth interviews were conducted using and an in-depth interview guide to collect data among females living with FGS (n = 10), females living with HIV (n = 3), female household heads (n = 10), community health management committee (CHMC) members (n = 6) and community leaders/opinion leaders. An IDI guide was used to explore their knowledge of HIV and FGS and care provision, as well as their views on integration and the facilitators and barriers of the integration of FGS into HIV care.

All data collection guides were designed in English and then translated into Ga, Twi, and Ewe. The translations were done by language experts using a back-to-back strategy. With this strategy, a language expert proficient in English and any of the local languages first translated the interview guide from the English language to the local language. Another language expert then retranslated the guide from the local language back to English and the two versions compared and harmonized. Where there were disagreements, these were discussed by the two language experts with a third language expert as a mediator. This was to ensure uniformity in the data collection tools.

Four graduate research assistants who are proficient in the local languages and skilled in collecting qualitative data were recruited and trained to assist in the data collection. The training was a combination of classroom work and mock interview exercises in both English and the local languages after which they were deployed to their linguistically competent areas to collect the data. The RAs were supervised by the investigators during fieldwork. The interview sessions for KIIs and IDIs lasted between 40 minutes to 1 hour, whereas FGD lasted for about two hours. Interviews were conducted in English, Ewe, Akan and Ga depending on the participants language proficiency. The study was conducted between January and April 2024.

## Data processing and analysis

All audio-recorded data were transcribed verbatim. Data collected in local language was first translated into English by two independent people. The translations were then compared for consistency. Any inconsistency was discussed by the translators with a third person serving as a mediator. Qualitative narrative data in English was then entered into a word processor

(Microsoft Word) and imported in a format that allows coding of the interview transcripts in QSR NVivo 13 software for analysis, a computer programme for textual analysis of large qualitative data sets. A codebook was developed, discussed, and accepted by the research team and used for coding the data. The data was analyzed by the research team. This study employed thematic content analysis as the primary approach to analyze the qualitative data. The research team began by conducting an initial, in-depth reading of all interview transcripts to gain a comprehensive understanding of the content and identify preliminary patterns and emerging themes. This process allowed the team to generate a set of key thematic areas that formed the foundation for the development of a codebook. The codebook was created through a collaborative process, wherein the initial themes were discussed, refined, and operationalized into specific codes. Each code was accompanied by a clear definition, inclusion and exclusion criteria, and illustrative examples to ensure consistency in its application across the dataset. Once the draft codebook was completed, it was reviewed and finalized through consensus among all members of the research team to ensure shared understanding and agreement on the coding framework.

Using the finalized codebook, the team proceeded to code the data systematically. The initial phase of coding involved assigning segments of the text to "free nodes" in NVivo, which allowed for flexibility and the emergence of new sub-themes. As the analysis progressed, these free nodes were further organized and refined into hierarchical "tree nodes," enabling the researchers to structure the themes in a coherent and meaningful way.

To enhance analytical depth and ensure contextual relevance, the data was classified according to the source characteristics. Each interview was linked in NVivo to specific attributes, including the geographical location of the respondent, their sex, and the setting in which the interview was conducted. This classification enabled the team to examine thematic patterns across different demographic and contextual variables, thereby enriching the interpretation of findings.

## Results

The results are presented under the following themes: knowledge about HIV and FGS, Care provision for HIV and FGS care, integrating care provision for FGS and HIV/AIDS and Perceived implementation facilitators/enablers.

### Socio-demographic characteristics of study participants

As shown in Table 1 below, a total of 52 respondents participated in this study. Majority of the study participants were females (80.8%) as compared to males (19.2%). Many of the study participants were between the ages of 30–39 years (34.6%) and 50–70 years (32.7%) and more than half (63.5%) were married. All key informants and Community Health Officers attained tertiary level education.

### Knowledge about HIV and AIDS

Majority of FGS patients and Community members have good knowledge of HIV/AIDs. Most of these participants expressed that, HIV is transmitted through blood with an infected person, and when a person encounters sharp objects used by infected persons. They also mentioned there is medication available that can manage the condition and prevent mother-to-child transmission among infected pregnant women. There was also a good knowledge about HIV/AIDs among all participants who were HIV patients.

> *"The disease is a blood disease, if someone uses a sharp blade and he or she has the disease and you go take the same blade and mistakenly, you get cut by this sharp object you can contract the disease. This disease is also contracted through sex. When an individual has this disease HIV and is not known and he or she has sex with another without protection the other opponent can easily contract the disease"* **(IDI, R5, Female Household)**.

> *"HIV is transmitted through blood, and it will affect your weight, however there is medication for HIV and when you take it, you will live longer and it can also prevent mother-to-child transmission"* **(IDI, R1, FGS Patient, Female)**.

**Table 1. Socio-demographics of study participants.**

| Characteristics of participants | Number of participants | | | |
|---|---|---|---|---|
| | KIIs | FGD | IDIs | Total |
| Sex | | | | |
| Male | 4 | 1 | 5 | 10 (19.2%) |
| Female | 9 | 8 | 25 | 42 (80.8%) |
| Age (years) | | | | |
| <20 | 0 | 0 | 0 | 0 (0) |
| 20-29 | 0 | 2 | 3 | 5 (9.6%) |
| 30-39 | 0 | 7 | 11 | 18 (34.6%) |
| 40-49 | 6 | 0 | 6 | 12 (23.1%) |
| 50+ | 7 | 0 | 10 | 17 (32.7%) |
| Educational level | | | | |
| No formal Education | 0 | 0 | 6 | 6 (11.5%) |
| Primary/Junior High School (JHS) | 0 | 0 | 17 | 17 (32.7%) |
| Middle School/Secondary/Senior High School (SHS) | 0 | 0 | 5 | 5 (9.6%) |
| Tertiary | 13 | 9 | 2 | 24 (46.2%) |
| Marital status | | | | |
| Single | 0 | 2 | 7 | 9 (17.3%) |
| Cohabiting | 0 | 0 | 1 | 1 (1.9%) |
| Married | 13 | 6 | 14 | 33 (63.5%) |
| Divorced/Separated | 0 | 0 | 2 | 2 (3.8%) |
| Widow | 0 | 1 | 6 | 7 (13.5%) |
| Total | **13** | **9** | **30** | **52 (100%)** |

Regarding participants' source of knowledge about HIV and AIDS, majority of community members mentioned that they first heard about HIV and AIDS through the radio and television. Few added that they got to know about it at the hospitals, and at the community level through education by health workers and announcements on the Community Information Centre. However, among FGS and HIV patients, majority of them got to know about HIV at the hospital either through antenatal care or visit for healthcare. This was followed by radio and television as other sources of HIV and AIDSa information, and a few mentioned the church, community and school as places they got to know about HIV/AIDs.

*"I hear it from the television, from the radio stations, they talk about it on the radio stations. The community centers also announce and talk about diseases to us" (IDI, R2, Female Household).*

*"At the hospital. They educate us on HIV and AIDS at the hospital anytime we go to antenatal clinics. They tell us not to be scared and encourage us to check our HIV status. They encourage us that there are medications for managing the HIV condition and early detection is important to avoid mother to child infection transmission" (IDI, R1, FGS Patient, Female).*

In addition, participants expressed that the information regarding HIV should be made available to the public through various media. The majority indicated door-to-door/home visits, use of community information centers, durbars, making information available in different languages and broadcasted through radio and television so that everyone can understand as well as through churches. These are evident in some participants' quotes below:

*"We can relay information about these diseases through information centres in various communities, holding durbars to talk about the disease to the people, through television and with different dialect because we can communicate to people but if we don't not communicate with them with the language they understand, we will not get the right people to get it and follow the information and go by it please. We can also relay information through the church to also create awareness"* **(IDI, R6, CHMC, Male)**.

*"If the information can be relayed through radios, televisions, at the marketplace, bus stations churches and beer bars to target the young who go there and schools such as the junior high schools and the university that could really help"* **(IDI, R3, HIV Patient, Female)**.

Knowledge about HIV and AIDs among community members and FGS patients have helped them to protect themselves against the virus by avoiding unprotected sex, not having multiple sexual partners as well as not sharing sharp objects with other people. They also mentioned that their knowledge about HIV and AIDs has encouraged them to periodically check their HIV status.

*"It has helped to protect myself from getting HIV, for example when I'm in a relationship with you or you have a boyfriend or girlfriend, you don't have to have sex with the person without knowing their HIV status, you two have to go to the hospital and know each other's status before you start having sex"* **(IDI, R2, Female Household)**.

Also, HIV patients expressed that, the knowledge they have received about HIV especially on the availability of medication and that, adhering to this medication will help them live healthier has given them some form of comfort. Also, the counsel they receive has helped them to take better care of themselves by eating well and avoiding unprotected sex, and educate others about the virus.

*"It's been very useful to me and I advise people I meet here, in Ghana, when you are living with HIV people stigmatize against you so I see these people here as a family… from the beginning, I was in a confused state but as time went on I encouraged myself that…when I went to see the doctor for consultation, he put me on medication so I asked him about the side effects of the medicine, things I'm supposed to eat and what not to eat, so he told me that I should stay off alcohol if I want to stay healthy and live long, I should also try as much as possible to avoid stress and I should exercise, and make sure I eat a balanced diet, and also I shouldn't take things that contain caffeine and acidic fruits and when I adhered to all these things, I realized it has been of help to me… The disease makes you weak but when I put on medication and adhering to all directives, I've realized that my strength is back to normal and I can do everything"* **(IDI, R1, HIV Patient, Female)**.

Participants also mentioned some preventive measures for HIV/AIDs. These include avoiding having multiple sexual partners, abstinence, if possible, use of condom, and avoid sharing sharp objects with other individuals regardless of your relationship with them.

*"We should take very good care of ourselves when it comes to our sex life. We must be very careful because you may not know if the person has the disease or not. I would advise that protection is used. In addition, we must avoid using sharp objects like blades even toothbrush with people and even with our family members because we may not know who really has the disease"* **(IDI, R5, CHMC, Female)**.

*"We have to protect ourselves from having multiple sex partners and also protect ourselves with condoms and if we can abstain too, it will help a lot. We should avoid using sharing some objects like blades, razor blades even with our relatives because we do not know who has the disease"* **(IDI, R8, FGS Patient, Female)**.

**Knowledge about female genital schistosomiasis (FGS)**

The study further explored participants' knowledge about FGS. Findings revealed that, many FGS patients and community members knowledge about FGS were mainly about experienced symptoms such blood in urine, itchiness and pain around the vagina and, that the disease is contracted from the river which is used by community members. For instance, some participants mentioned that:

*"It is about the river. The are many sicknesses in the river. When we spent much time in the river… These are small germs in the river. Even when you drop a piece of wood in the river, after some time, you will realize small snails like creatures on the stick. We get it from the river. Anytime we spent time in the river, which is when we are infected…I have taken medications to treat this condition several times. The doctor told me that the germs passed through my genitals into my womb. Anytime I urinate, I see blood samples. It was confirmed after a lab test. They prescribed some medications for me…As a result, I stopped fetching water from the river"* **(IDI, R2, FGS Patient, Female)**.

*"I know is a disease contracted through the water. In this community, we do not have pipe water or a borehole so all we use is the river water so we easily can contract it. I also know that when you are infested with disease, you can find blood in your urine, you will have itches with your body and have sores around your vagina as a woman"* **(IDI, R6, Female Household)**.

It was however realized that all HIV patients who participated in the study do not have much knowledge about FGS. The following illustrate this point:

*"I've heard about it before, but I don't know anything about it"* **(IDI, R1, HIV Patient, Female)**.

*"No, I don't know about it"* **(IDI, R2, HIV Patient, Female)**.

*"I don't really know about this disease, all I know is that you can contract it through water bodies please"* **(IDI, R3, HIV Patient, Female)**.

Furthermore, the study revealed that, key informants have an in-depth knowledge about female genital schistosomiasis:

*"So, Schistosomiasis is a disease of the blood fluid. Usually, the organisms. Schistosomiasis affects the blood, the tracts, the GTI and the eggs. When the parasite attacks the human through the skin, it passes through the skin, and it enters the blood. And when it goes, it lays eggs. And the eggs are excreted through the urine and the feces of the human. So, once the eggs are laid and it comes to a bladder and urinates in the fresh water, it contaminates the water. Or passes through the water, but it contaminates the water. And we have water snails, small, small water snails, which also ingest the parasites. And the developmental process also goes in the waters, and they also release it into the water. Once you are walking or playing, swimming, or doing anything in the water, the parasite pierces your skin and enters your system"* **(KII, R2, Male)**.

Community Health Officers (CHOs) however expressed that, they do not receive any specific training regarding FGS and that they are able to identify FGS based on the symptoms presented by the patients which they sometimes associate with bilharzia hence, they often refer to such patients for laboratory test and further diagnosis.

*(Interviewer: Please, per our training, do we know how to identify a person with Schistosomiasis?): "All Respondents: NO"* **(FGD, CHOs)**.

*"You see, with the FGS, we received some training from Unilever Company, I did join, and we also did some at our facility. As for us the CHOS the only problem is that when we suspect somebody for FGS per our investigation and*

*findings, we normally refer them to the facility so that lab investigation can be carried on upon them…Sometimes with the FGS we associate it with bilharzia so normally for me; I do refer people to come to Amanfrom for lab investigation to rule out the specific disease whether it be FGS or an STI"* **(FGD, R2, CHOs)**.

*"Is the same as my sister said we refer them, especially with female with blood in urine. Seeing clots in their blood gives me a preview and also a severe abdominal pain. So due to this, I ask them to go to the health facility"* **(FGD, R3, CHOs)**.

The study further explored the sources of knowledge of FGS and the usefulness of this knowledge among study participants.

Many of the female community households mentioned that there learnt about FGS through education provided by some researchers from the University of Ghana while few of them mentioned they heard of FGS from community awareness creation by doctors and nurses from facilities within the communities while others also learnt of the disease due to experience of contracting FGS. These are expressed in some participants' quotes below:

*"Certain doctors came from Legon Noguchi, to come and talk to come and educate, teach and took some blood samples from us, left then after came in a months' time and brought us medications to treat the disease. That is the FGS. Sometimes too, they move from house to house"* **(IDI, R7, Female Household)**.

*"I heard it from nurses and some health programs were organized in the community and we had some health professionals coming in to talk about this disease (FGS) with us"* **(IDI, R9, Female Household)**.

About half the number of FGS study participants indicated they got to know about FGS after they experienced the symptoms and sought medical care while the remaining half said they received education from health workers who came to educate the community about FGS. Below are some quotes shared by FGS participants:

*"It is a personal experience. As I told you earlier, these conditions commonly experienced by the community members because we depend on the river for everything"* **(IDI, R1, FGS Patient, Female)**.

*"The nurses came here to tell us about it, they told us we can get sickness from the river, so we should avoid bathing and swimming in it"* **(IDI, R6, FGS Patient, Female)**.

*"Some health professionals came around to talk about this disease to us and even supplied medications to those who have the disease or do not have it"* **(IDI, R8, FGS Patient, Female)**.

Regarding the usefulness of the knowledge received, almost all community and FGS patients expressed that, the awareness they received on FGS has influenced them and their family to stop bathing and swimming in the river. Others also stopped drinking from the river and in situations where they will use the water from the river to cook or bath at home, they will either allow it to settle, add alum to the water to purify it or boil it to kill the living organisms:

*"I have stopped my children from swimming or bathing in the river, they can go the shore, but they should desist from swimming inside the river. sometimes when you fetch the water and bring it home, you see some living organisms inside, so you have to put in alum for it to settle or boil the water before we use it"* **(IDI, R3, Female Household)**.

*"I had to prevent my young girls from playing in the water since they used to do that a lot. Further, I decided to boil water before usage. Now, in my household, we drink sachet of water that is the one they sell"* **(IDI, R8, Female Household)**.

*"I don't allow my kids to take their bath at the river side anymore, they fetch it home and bath. When I fetch the water and bring it home, I boil or put in alum for it to settle and then I use it"* **(IDI, R5, FGS Patient, Female)**.

The study further explored other sources from which participants will want to receive information related to FGS. Many of the participants expressed that information about FGS can be communicated through the radio, television, community information centers and churches. Others also mentioned information can be shared at hospitals through Child Welfare Clinics, marketplaces, lorry stations and during community durbars. Some community members and patients expressed that:

*"You see with this disease; I had no knowledge about it until these personnel's came around to give us this information. So, I think if we also relay it through the FM stations, when we go to the clinics nearby, they talk more about it and make some advertisements. When durbars are held, health professionals can come around and enlighten us more about the disease and continue to caution us"* **(IDI, R5, Female Household)**.

*"I think we have our child welfare clinic in the community, I think when the mothers go there, health workers can take advantage of that to communicate with the mothers about the FGS disease so that the mothers can take care of themselves and their children"* **(IDI, R5, CHMC, Female)**.

*"I was able to talk to my friends about the disease. I think, when it goes on the radio, we go from house to house, also through screenings that are held during community durbars and also through our churches. Most people go to church, so I know when information is relayed through the church it will get to most of the community members to take care of their family and themselves as well"* **(IDI, R9, FGS Patient, Female)**.

*"Information about this disease has to go viral. Can you believe this? I think if they pass on information through radios, television, holding screenings, talking about it during grand durbars, nurses coming to our communities to hold health programs to talk about this, in my opinion, it will be great. Doing all this, you can reach a lot of people and as I said, I don't really know about this disease so as to my other community members wouldn't know or other people wouldn't really know about this for them to be cautious"* **(IDI, R3, HIV Patient, Female)**.

When it comes to preventive measures for FGS, many of the FGS patients mentioned treating the water by boiling or by adding alum, avoiding bathing or staying longer in the river and said there is medication available to treat FGS. However, among community members, in addition to treating the water fetched from the river by boiling or adding alum as well as not bathing in the river and creating awareness, they indicated that, the best preventive measure for FGS is providing the community with pipe-borne water. These are expressed in some of the illustrative quotes:

*"There is a medication to treat FGS. The moment it is administered…the medication kills all the causative agents in your system. There is medication for FGS. They distribute to us periodically"* **(IDI, R1, FGS Patient, Female)**.

*"The only means for us is the provision of pipe borne water. This is because, no matter the frequent distribution of medications for FGS, the sickness will still be with us because we do not have any other alternative than to depend on the same water causing the sickness. It all depends on the alternative water source"* **(IDI, Community Leader, Male)**.

*"I know of boiling water before usage, not swimming in the water, you can use alum too. In all, I would say that the community should be provided with pipe water so that all of this will come to an end with the FGS. This is because, the disease is contracted through the water and the more we enter to fetch we will still contract the disease so building us a pipe for the community, will even stop us from entering the water and that prevention will be a permanent one for us"* **(IDI, R5, Female Household)**.

*"Hmm! I know of heating up our water before, not bathing in the water, also taking in the medication that is brought up by the health workers to protect you from the disease. In all these, we do a lot of things with water, and I think, water is a necessity and for us to really prevent this we will opt for pipe borne water in our community. This is because, we go in to fetch this water before we even come to boil it is used for washing and in all these activities you will always contract the disease so I will suggest the pipe borne water that will prevent us from entering the water, we won't take medications anymore and everybody will live happily"* **(IDI, R7, Female Household)**.

### Care provision for HIV and AIDs

Participants from the various categories mentioned that care provision for HIV is mainly sought from the hospital or health centers. Most community members expressed that, due to enlightenment and awareness creation about HIV and AIDs, most people seek care at health facilities though few others still visit traditional herbal places and churches. This is evident in some community members' quotes:

*"To be honest with you, education on HIV and AIDS has enlightened many on the need to visit the hospital or health centre for HIV and AIDS care. To be honest, some also visit the herbal centres"* **(IDI, R1, CHMC, Male)**.

*"Let me use my mother's health issue as a case, when my mother was diagnosed with HIV, she thought that was the end of her life. Many people advised her to seek care from spiritualists and herbalists. Many people roam seeking care. However, I think hospitals are the best place to seek care. We advise our mother to seek care from the hospital. She refused"* **(IDI, R2, CHMC, Male)**.

*"Others also seek care from herbal practitioners. But majority seek care from the hospital for them to determine the viral load and how the condition can be managed. There are medications to manage HIV and AIDS"* **(IDI, R1, FGS Patient, Female)**.

In addition, other participants elaborated that, patients travel to other regions or communities to seek HIV care due to issues of stigma:

*"For me, I come to (facility name) polyclinic for care I was literally transferred from (Another facility name). So, from then, I have been seeking care…People do seek for care at other polyclinics like Kasoa polyclinic, people go all the way to Winneba because they do not want others to know of their status. And sometimes the herbal clinics but I prefer here"* **(IDI, R3, HIV Patient, Female)**.

*"…Usually, it's also for the individual to determine where he or she is comfortable accessing services. There are some people who reside in Adabraka but they prefer going to Dodowa District Hospital to access services because of stigma. Some people attribute HIV to being a spiritual problem. So, once their intention is spiritual, they will never come to a health facility"* **(KII, R2, Male)**.

The findings also revealed challenges that are associated with health care facilities where HIV care is provided. The most common challenges realized were the shortage of medication, inadequate logistics and health care providers.

*"When it comes to the (facility name) polyclinic their approach of health care is really good, there are no delays, and they also do really take care of emergencies. But unfortunately, the beds there for patients are not enough, they do run out of medications quickly and lack some logistics to carry on some test and operations…there need to be provision of more facilities, and more trained nurses and doctors"* **(IDI, R6, CHMC, Male)**.

*"If I would say (Facility name) should have drugs always to prevent shortage more also Kasoa will be provided with more working staff like nurses and a laboratory with testing kits. With Winneba, I cannot say much about it because I have been there and is quiet far from my place"* **(IDI, R3, HIV Patient, Female)**.

## Care provision for female genital schistosomiasis

Regarding where community members seek care for Female Genital Schistosomiasis, many of the participants expressed that, care is usually sought from health facilities within the community.

*"We visit the hospital for care… We only seek care from the hospital"* **(IDI, R1, FGS Patient, Female)**.

*"Some health professionals came around the community to take care of us having the disease. That is, bringing us medications every month. Some members go to our nearby clinic and Amanfrom polyclinic"* **(IDI, R9, FGS Patient, Female)**.

Some community members however indicated that, initially, many people perceive it as a spiritual sickness as it was associated with the river hence, they seek traditional and spiritual treatments.

*"Individuals do seek care at different places. At first, we didn't really know of the disease so some symptoms like blood in the urine and having itching body, after the use of the water, some thought it was a spiritual sickness because of the water people use, so most people were resorting to herbal clinics and the spiritualist"* **(IDI, R5, Female Household)**.

*"Some go to the herbal clinics and churches. Churches because is a water disease they assume or take it to be a spiritual sickness, so they prefer going to the church to be prayed for"* **(IDI, R6, CHMC, Male)**.

The study further revealed that there are challenges with identifying FGS cases, diagnosis and recording especially among healthcare providers. Most of these challenges were associated with unavailability to test and diagnose FGS at the community level hence, persons who present symptoms like FGS are either referred to higher facility for laboratory test, and some of these people end up going for herbal treatment. Also, due to FGS having similar symptoms as STIs, they are mostly recorded under STIs within health records hence it is difficult to get data on the number of FGS cases from health facility records within the community though high number of people are presenting cases of FGS. Below are some quotes from key informants highlighting challenges with FGS diagnosis and recording:

*"For our place, Tomefa, usually when we share or talk about the disease to them, they know where they fall but normally, they come with severe urinary tract infection When they come, we do a first line test with them by going to the laboratory to be tested. We refer them to Amanfrom Polyclinic but normally if you refer them, they wouldn't come. Instead, they tend to visit these herbal shops to buy some drugs over there rather. So, at our place we do not have any specific thing used in identifying the female schistosomiasis. We have a known procedure for treating the disease. So, we normally treat it as a urinary tract infection"* **(FGD, R4, CHOs)**.

*"…That it isn't captured. It's probably handled as an STI and the person is given some medicines, and it is tallied as genital ulcer or urethral discharge or something, which is an STI. So probably that is what happens until they go to meet a clinician who is taking a proper history to know your background, to even suspect that because you've had exposure to certain environments that have certain water bodies, you are likely to, this could be a differential and then they chart that path. So, it is unfortunate. It tells us that we have to really open our eyes"* **(KII, R1, Female)**.

*"Anything that is genital infections are STIs. So, we always classify it as an STI. So, I have recently made a proposal to train clinicians. Especially last year, when we were about to have the campaign, the exercise. Through the discussion, we were informed (name of Health worker) that there can be organized training for the clinicians. Okay. So, they can also be aware that we have female genital schistosomiasis, and it will help us a lot…. Yes. If the clinician suspects it as bilharzia or schistosomiasis, then records will also document it as such. But if the clinician doesn't suspect it's an STI, it will be difficult for you to give me documentation to also change the STI to a schistosomiasis"* **(KII, R2, Male)**.

Some possible solution suggested to address challenges with FGS diagnosis and record keeping at the healthcare facility levels include training of health care providers and awareness creation:

*"We have to organize training at all levels so that we will have knowledge about the condition and where to refer and how to treat it, because we have health centers, we have CHPS compound, so at the CHPS compound, they have to get more knowledge about it so that when the cases come, they will know what to do"* **(KII, R3, Female)**.

*"In my opinion, I think staff at the various clinics and hospital must be well trained to elaborate more on the disease FGS, to be able to provide quality health care for the patients"* **(KII, R6, Male)**.

### Integrating care provision for female genital schistosomiasis and HIV and AIDS care in the community

The study participants had divergent views regarding the integration of care provision for Female Genital Schistosomiasis (FGS) and HIV and AIDS. Some community participants were not in support of integrated care for FGS and HIV and AIDS. They cited cultural and societal factors such as norms, beliefs, and practices related to both diseases, along with concerns about discrimination and stigma associated with HIV and AIDS. Participants also raised issues regarding treatment methods and diagnoses.

*"I don't think integration of this disease care will be possible. This is because a lot of people do talk about it and these two people going to the same place will be assumed in the minds of people that all have contracted the HIV disease. So, me for example, if I have been diagnosed for FGS and am told to go to the same place where HIV patients go, I wouldn't"* **(IDI, R7, Female Household)**.

*"It is not good… it is not good at all. Clients may be misdiagnosed. In addition, it will promote stigmatization as well as infringing on health privacy. People may spread health information of client"* **(IDI, Community Leader, Male)**.

*"With the integration of both the FGS and HIV and AIDS, I don't think Integrating their care together will be a good one, this is because of stigmatization. For instance, if I am battling with FGS, I don't think, I will go to such a place for care because I wouldn't feel comfortable. Many thoughts will run through my head and people may stigmatize me…I will not like the integrated care because, humans can make mistakes and health workers could mistakenly give HIV medication to FGS patients and vice versa when persons having these separate conditions are receiving care from the same unit. So, I will prefer separate care for FGS patients and HIV patients"* **(IDI, R5, Female Household)**.

Furthermore, most FGS patients fear they will be associated with HIV and AIDs within the community hence will not want to receive treatment where HIV care is given:

*"I do not agree on this at all. What will people say? And for instance, I being an FGS patient, and you tell me to patronize my health care at the same place where HIV patients go, I wouldn't go at all. This is because people talk too much; I am going to be seen in a different light which I wouldn't like. They will even classify me as part of the HIV patient"* **(IDI, R8, FGS Patient, Female)**.

*"For that I do not think I will agree to this. What would people say if for instance they see me coming from a pace like that? It is not a bad place, but people will begin to think someway of you…for me it won't be accepted at all because of how people talk and even how people will begin to act towards you"* **(IDI, R10, FGS Patient, Female)**.

Though some HIV patients were okay with the integration, others also were not in support due to issues of stigma:

*"I think it's not good, you see I told you that even when I came here, I was very skeptical about whom to even ask for directions to the HIV unit, I can't just ask any random person and they will stigmatize against me, so I don't think they should be mixed, we are all human beings but I don't think we should be mixed… you know we the HIV patients and that of the FGS patients are not equal, so, in case we all meet at the same place they might go and tell on you or stigmatize against you, because when you have malaria, you have a place you will go to, if you have a mental situation too same, so it should be the same for HIV patients too, we shouldn't be mixed together, everybody should be in their lane"* **(IDI, R2, HIV Patient, Female)**.

Conversely, many key informants viewed integration of FGS and HIV positively, especially for HIV clients who may have FGS. They highlighted integration as a cost-effective approach that will reduce stigma, promote holistic patient care, and optimize resource utilization. However, there must be a lot of capacity building for health care providers and availability of logistics to enable effective integration.

*"Okay, so integration is the way forward. I mean, once you integrate, you build capacity so that this service provider, wherever they find themselves, will be able to provide care. And integration is economical. It makes a lot of sense to integrate so that you don't go to the hospital one day for your HIV care and then go another day for your other condition, for this other care, for this NTD [Neglected Tropical Disease] that you have. Now, we need to integrate as much as possible. Get people sensitized. So, if there's the service provider who is already providing ART [antiretroviral treatment] care, why don't we just provide some capacity building, build their capacity in how to manage FGS"* **(KII, R1, Female)**.

*"Yeah, the reason as I said, it will be cost effective. Already people who have HIV and AIDS don't want to be seen to be going around and all. They want to maintain confidentiality and things. So, if the people managing them, the HIV, also know that they have the FGS, then they will be able to manage that in addition to other things that they are managing. So, I recommend the integration of those two services. They are key"* **(KII, R8, Male)**.

**Perceived implementation facilitators and enablers**

Perceived implementation facilitators/enablers were also explored among study participants at the individual, community and facility levels.

At the individual level, the most common enabling factor for a successful buy-in to integration of FGS and HIV care was education and awareness creation. These are evident in the narratives below:

*"If education is done properly, I can even educate my friends and family as well"* **(IDI, R10, FGS Patient, Female)**.

*"If I gain the understanding of why the integration is taken place and how carefully it will be done, if the health professionals are going to put things in place like offering their people good training on this integration and how they will treat and take care of things like medication, then I will also inform and explain to my friends about it and my family and neighbours about the initiative.* **(IDI, R9, Female Household)**.

*"Education, at the individual level. It can be individual, at school, the churches, and communities. Education is the key to all this. Because if you don't understand the thing, how will you accept the person, or how will you deal with the person? So, education is the key factor"* **(KII, R13, Female)**.

In addition to the above, some other participants indicated that, reduction in stigmatization, confidentiality of clients (patients) health records and effective insurance will also facilitate buy-in of FGS and HIV care integration at the individual level. These are expressed in some participants' comments below:

*"At the individual level I see reduction in stigma as a facilitator or enabler that will enable people to accept the integration. Another facilitator at the individual level will be efficient health insurance. If the health insurance works properly that will be an enabler towards the integration of care"* **(KII, R1, Female)**.

*"Trust must be built between the health workers and the clients. In addition, confidentiality must be assured. If the clients later hear of its health information in the public, it will affect the integration drive"* **(KII, R4, Female)**.

At the community level, enabling factors identified for an effective care integration for FGS and HIV care include engagement of key stakeholders such as opinion leaders, chiefs, Assembly members throughout the planning and implementation processes as well as education of community members through mediums such as community durbars and health screening activities. Below are some statements from study participants:

*"They must organize a community durbar. The chiefs, the assembly members, unit committee members and the entire community must be educated on the integration. In addition, there will be the need to use the community health volunteers to educate the community. In addition, the public address system in the community must be utilized to send information on the integration across"* **(IDI, R1, CHMC, Male)**.

*"I think the community must be well communicated about this integration through durbars, coming to churches stating the advantages and disadvantages about this integration"* **(IDI, R5, Female Household)**.

*"We create awareness through durbars and opportunity that will get us into the community. We can use information centres, go to churches, screenings held at marketplaces and bus stations. This is because; screening brings in a lot of people. If their sugar level is being checked, being educated on their diet, we can also add information about the FGS too. For instance, I would say last month, we carried on some screening in the community, and we added on service for registering individuals for their National health Insurance Scheme (NHIS). And we had a lot of people trooping in. So, with this we can add on service with the FGS, and it will be effective"* **(KII, R6, Male)**.

*"And the community, I think you people should involve the opinion leaders in the community too. Wherever you go, I think if you go to a community and you involve the opinion leaders, and you let the opinion leaders, you educate them, you let them talk to the people in their own language, and they understand it better than maybe a stranger going to talk to them"* (KII, R13, Female).

In addition to the above, some participants stressed on the importance of continuous awareness creation on HIV among community members to decrease stigmatization of Persons Living with HIV to enable an effective integration of FGS and HIV care at the community level:

*"The community needs to be educated more about HIV and FGS, there is this lady who told people about her status in the community, and the kind of treatment they give her, I am always sad when I see her, but you can't say much because it's their family affair so the community needs to know that though you are HIV positive you are human just*

*like any other person so the community really needs to be educated… I think it's all about the communication and education, if you don't educate them, the integration will not be successful so when the community accepts everybody irrespective of your status it will be okay"* **(IDI, R1, HIV Patient, Female)**.

For the facility level, the study revealed many perceived enabling factors for an effective integration of FGS and HIV care. Some of the major enabling factors identified include the availability of logistics, medication and the accessibility of these mediations by patients as well as the quality of health care provided:

*"It is about the availability of medications and the diagnostic tools"* **(IDI, R2, FGS Patient, Female)**.

*"If at a national level we don't have the medicines for treating the condition, yes, I've been tested, it's been seen, but the medicines to manage the condition is not there. So, I'll still be going round, round and round. If the medicine is there, it is expensive. I cannot pay for it also. It is there, but I cannot afford it. So, I am also denied access to it. So, these are the things. Then the quality of the service itself. You get it and they don't even tell you how to use it properly so"* **(KII, R8, Male)**.

*"The government should also play its part. It should give you people whatever you need for the work to go on. Supply you the drugs, the personnel should be there, everything should be there. Because of all these things, if you don't have the resources, I'm sorry, how do you do the work? So, the government should also play his part"* **(KII, R13, Female)**.

Participants also stressed on training and capacity building for health care providers to enable them effectively to provide integrated care for FGS and HIV:

*"There should be periodic training. WHO always comes up with new ideas and people clinicians and staff need to be trained on these issues. I think health workers will need training and retraining. Health space as well. It's all part of the conditions of the health worker"* **(KII, R1, Female)**.

*"At the facility level, more trained health workers must be provided, more logistics and medications must be provided to help facilitate the integration properly"* **(IDI, R6, Female Household)**.

Some other study participants also mentioned the need to utilise existing HIV structures and the need to provide some motivation to healthcare providers to enable an effective FGS and HIV care integration at the facility level. These are expressed in the participants' comments below:

*"Okay, so for HIV, the system has already been established and that is a potential enabler for integration. We are just adding on the service. HIV already exists, we are testing, educating, managing, counselling and treating. Since the structure has already been established, it will be easier for us to pluck the FGS service"* **(KII, R6, Male)**.

*"We should motivate our health workers. The nurses and the doctors should be well motivated so that they will give them (patients) the best care when they come there. In terms of counseling, in terms of handling, in terms of all the services that they provide. If they are well motivated, they will do the work as expected. And it will encourage people to be coming to the facility all the time"* **(KII, R12, Male)**.

## Discussions

The study found that knowledge on HIV was high among all study participants as compared to FGS. Community members were able to describe the mode of transmission for HIV, the availability of HIV treatment/medication and prevention methods for HIV and how this knowledge has helped many community members protect themselves from HIV/AIDS infection. This high level of HIV/AIDs knowledge could be attributed to the increase awareness creation of HIV over the years which has also been realized in other studies as a widely known public health problem among diverse population [22].

Knowledge of FGS was however realized to be generally low especially among community health workers and HIV patients. Most female community members who expressed knowledge about FGS described gynecological symptoms experienced by them which include blood in urine, itchiness and abdominal pain and pain around the vagina [10] and they mostly refer to the disease as water-borne disease. All participants indicated the use of the river within the community was the mode of transmission of FGS.

The knowledge gap of FGS identified in this study is evident in other studies conducted in Ghana and other African countries that also realized low level of FGS knowledge among community members and health workers [23–27]. The low level of knowledge of FGS among study participants especially Community Health Officers could be attributed to the disease not being mentioned in medical textbooks or thought during health trainings, or limited awareness creation and education on FGS at community level [23,28–30]. There is therefore the need for an increased awareness creation of FGS among community members through mediums such as the radio, television, community information centers, churches, Child Welfare Clinics (CWC), marketplaces, lorry stations and community durbars as indicated by study participants. Also, there need to be an intensive training of health workers especially Community Health Officers working in Schistosomiasis endemic communities, on Female Genital Schistosomiasis to address the knowledge gap of FGS.

Regarding care provision for FGS and HIV, many of the study participants indicated that care is mainly sought from hospitals, health centres and clinics. It was also realized that, some HIV patients have to travel far to seek HIV care at farther places often in another district or region due to issues of stigma [31–33]. Though there has been progress made regarding HIV care in Ghana over the years, there is still the need to address challenges such as stigma, availability of resources needed for HIV treatment and increased coverage of Antiretroviral Therapy ART [29].

Generally, FGS was diagnosed and treated as STIs and this misdiagnosis has been reported as a common issue in FGS endemic communities in other African settings [23,34]. In Ghana the adoption of the syndromic management for STIs and lack of specialized health workers in the community negatively affected the early detection and management of FGS among people who have HIV. The inability of health care providers to adequately diagnose and treat FGS has been realized as a major gap in addressing FGS and co-morbidities [25]. This calls for the need to build the capacity of health workers on FGS, especially those who operate at the community level (Community Health Officers). When this is done, it will lead to better understanding and community surveillance for both HIV and FGS given the fact that Ghana is endemic country with distribution of the condition clustering around communities with water bodies. Communities in which the FGS infection is most endemic have limited access to clean water and healthcare services [31]. In such endemic communities it is estimated that between 33–75% of women have FGS yet few of these cases are diagnosed correctly [9]. Though some community members seek FGS care from health facilities, others still attribute FGS as a spiritual sickness as it is associated with the river and therefore seek traditional, herbal and spiritual treatment for FGS [23,35,36].

Ghana schistosomiasis control efforts are mostly relying on the Mass Drug Administration of Praziquantel to school aged children whilst neglecting other risk groups such as reproductive aged women. However, these efforts may not yield the desired impact if cases are not diagnosed early and treated. Undiagnosed cases in the community become the reservoir for infection [37]. Therefore, to break the cycle of infection would require treating infected individuals. This is important in the country drive towards the elimination of HIV through the achieving the 95-95-95 global agenda.

In terms of implementation of integrated model for HIV and FGS, participants living with FGS or HIV/AIDS, including community health committee members, female household participants, and some school health teachers, shared concerns about stigmatization and the mode of transmission, leading them to oppose integrating FGS and HIV care. Both HIV and FGS are associated with severe social stigma in Africa. There is the need for intensified education about both diseases at the community level. Stigma can be reduced through community-based screening of women during outreach. Both HIV and FGS screening can easily be done at the community level using Oroquick and urine dipstick for microhaematuria respectively [26]. To be able to effectively integrate these two conditions would require development of clinical

protocols for female genital schistosomiasis. In thus study, health workers revealed that the absence of clinical protocol as one of barriers for routine screening of women for FGS.

Despite the low knowledge of FGS, the study identifies some facilitators for effective implementation of integrated care for FGS and HIV/AIDs. Use of community engagement strategies such partnerships and collaborations between various outlets for service delivery was recommended. Participants acknowledge the existence of different categories of health care providers including biomedical, spiritual and herbal healers and care was sought from these outlets either sequentially or concurrently (Fig 2). In a plural medical systems, people may identify with one or more of the three health systems; professional sector (biomedical), folks and the popular system [34] and in recent times, spiritual sector. Both orthodox and alternative health systems exist in Ghana. Many individuals, especially those from non-Western cultures, have been reported to have a holistic concept of health and disease that entails a spiritual aspect of disease causation [38]. In Ghana, biomedical model of health and disease concepts are widespread. Nonetheless, earlier qualitative studies demonstrate that many people still hold holistic concepts of health that incorporate spiritual factors in addition to physical and psychosocial factors [35,36]. Government should establish effective partnership with community influencers such as chiefs, Imams, and religious leaders to quickly assess the community readiness and to facilitate any future interventions in the community in case of public health emergencies and tragedies. The low knowledge could continue to push people toward non-orthodox health outlets who may not have the capacity to detect and manage FGS.

In addition, increasing awareness creation regarding FGS to address existing knowledge gap and care as well as destigmatisation of HIV among community members are crucial for effective integration of FGS and HIV/AIDS. Consequently, health worker capacity needs to be built especially regarding FGS diagnosis and care as well as in the provision of an effective integrated care for FGS and HIV/AIDS. Provision and availability of needed logistics, human resources and medications for both FGS and HIV are also crucial for successful integration. Addressing these gaps will contribute to an effective and sustainable integrated care for FGS and HIV and AIDS [36].

While awareness creation and education among community members, capacity building of health care providers and administration of MDA will help control the incidence of FGS, effective solutions such as provision of portable drinking water within endemic communities are key in preventing FGS transmission in Ghana. Also, providing an integrated care for FGS and HIV and AIDs will contribute to a reduction in long-term reproductive health problems to women and girls who are often marginalized groups within communities. Local, regional, global health professionals and policy makers need to therefore play their respective roles in addressing existing gaps for FGS knowledge and care provision for FGS and HIV/AIDS to enable successful and sustainable integrated care for FGS and HIV/AIDS.

### Study limitations

Our study did not adequately elicit community members' knowledge on the association between freshwater snails, the intermediate host, their specific habitats and schistosomiasis. One of the core strategic interventions proposed in the

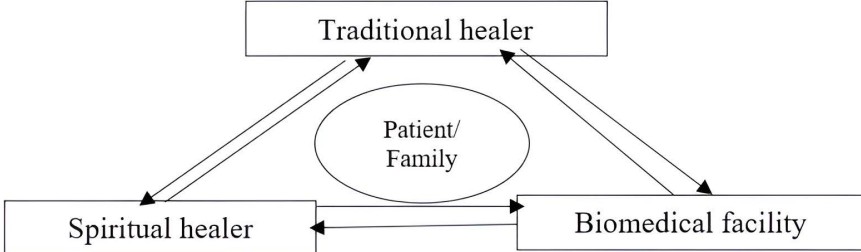

**Fig 2. Health seeking behaviour of people with FGS.**

WHO road map for neglected tropical diseases 2021–2030 for ending the neglect to attain the Sustainable Development Goals is snail control with molluscicides, physical removal and environmental modification [5]. It is imperative that community members are aware of these interventions and understand which role(s) they can undertake to contribute towards control/elimination efforts in their settings,

## Conclusions and recommendations

Although there was evidence of direct link between FGS and HIV, both community members and community health officers/workers had a generally poor level of knowledge about FGS. This was negatively affecting routine screening of females for genital schistosomiasis. Integration of FGS and HIV has the potential to contribute towards Ghana achieving elimination of HIV, however barriers in implementation need to be addressed before the roll out of such a program. The findings also suggest that developing partnerships and collaboration with different health care providers in the community may contribute to early detection and management for HIV and FGS. Finally, there is the urgent need to develop a national clinical protocol for integration and capacity building of community health workers. Further, there is a need to educate community members on the linkage between schistosomiasis and freshwater snails and the role they could play in the control and elimination of the disease intermediate hosts.

## Supporting information

**S1 File. Transcripts of focus group discussions and in-depth interviews.**
(DOCX)

## Acknowledgments

We would like to thank all stakeholders: members and staff from the Ghana Health Service (GHS), NTD and FGS health care professionals and providers at the National, Regional, District and community levels, the team of Ga South Municipal Health Directorate led by the District Health Director, Staff of Amanfro Polyclinic and all CHPS facilities within the study district, as well as FGS and HIV patients and community members who committed time to share experiences and provide data for this study. Also, we are very grateful to the field staff who participated in the data collection process for their meticulous work.

## Author contributions

**Conceptualization:** Emmanuel Asampong, Franklin N. Glozah, Adanna Nwameme, Ruby Hornuvo, Philip Teg-Nefaah Tabong.

**Formal analysis:** Adanna Nwameme.

**Funding acquisition:** Emmanuel Asampong.

**Investigation:** Emmanuel Asampong, Franklin N. Glozah, Adanna Nwameme, Edward Mberu Kamau, Philip Teg-Nefaah Tabong.

**Methodology:** Emmanuel Asampong, Franklin N. Glozah, Adanna Nwameme, Ruby Hornuvo, Edward Mberu Kamau, Philip Teg-Nefaah Tabong.

**Resources:** Emmanuel Asampong, Franklin N. Glozah, Edward Mberu Kamau, Philip Teg-Nefaah Tabong.

**Software:** Ruby Hornuvo.

**Supervision:** Emmanuel Asampong, Franklin N. Glozah.

**Validation:** Ruby Hornuvo.

**Writing – original draft:** Ruby Hornuvo, Philip Teg-Nefaah Tabong.

**Writing – review & editing:** Emmanuel Asampong, Franklin N. Glozah, Adanna Nwameme, Edward Mberu Kamau.

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
