## [Decision Letter · Decision Letter 0]

PNTD-D-24-01188Stakeholders Perspective of Integrating Female Genital Schistosomiasis into HIV Care: A Qualitative Study in GhanaPLOS Neglected Tropical Diseases Dear Dr. Tabong, Thank you for submitting your manuscript to PLOS Neglected Tropical Diseases. After careful consideration, we feel that it has merit but does not fully meet PLOS Neglected Tropical Diseases's publication criteria as it currently stands. Therefore, we invite you to submit a revised version of the manuscript that addresses the points raised during the review process. Please submit your revised manuscript within 30 days Jun 14 2025 11:59PM. If you will need more time than this to complete your revisions, please reply to this message or contact the journal office at plosntds@plos.org. Please include the following items when submitting your revised manuscript:* A rebuttal letter that responds to each point raised by the editor and reviewer(s). You should upload this letter as a separate file labeled 'Response to Reviewers '. This file does not need to include responses to any formatting updates and technical items listed in the 'Journal Requirements' section below.* A marked-up copy of your manuscript that highlights changes made to the original version. You should upload this as a separate file labeled 'Revised Manuscript with Track Changes '.* An unmarked version of your revised paper without tracked changes. You should upload this as a separate file labeled 'Manuscript '. If you would like to make changes to your financial disclosure, competing interests statement, or data availability statement, please make these updates within the submission form at the time of resubmission. Guidelines for resubmitting your figure files are available below the reviewer comments at the end of this letter. We look forward to receiving your revised manuscript. Kind regards, Hira L Nakhasi, Ph.D.Section EditorPLOS Neglected Tropical Diseases Hira NakhasiSection EditorPLOS Neglected Tropical Diseases

Shaden Kamhawi

co-Editor-in-Chief

Paul Brindley

co-Editor-in-Chief

 **Additional Editor Comments:** Please revise the manuscript based on the reviewers comments  **Journal Requirements:**

1) Please provide an Author Summary. This should appear in your manuscript between the Abstract (if applicable) and the Introduction, and should be 150-200 words long. The aim should be to make your findings accessible to a wide audience that includes both scientists and non-scientists. Sample summaries can be found on our website under Submission Guidelines:

2) In the online submission form, you indicated that "The data is available on request sent to the Administrator of Ghana Health Service Ethics Review Committee at ethics.research@ghs.gov.gh". All PLOS journals now require all data underlying the findings described in their manuscript to be freely available to other researchers, either

- In a public repository.

- Within the manuscript itself.

- Uploaded as supplementary information.

3) Please ensure that the funders and grant numbers match between the Financial Disclosure field and the Funding Information tab in your submission form. Note that the funders must be provided in the same order in both places as well.

 **Reviewers' comments:** Reviewer's Responses to Questions

**Key Review Criteria Required for Acceptance?**

**Methods**

-Are the objectives of the study clearly articulated with a clear testable hypothesis stated?

-Is the study design appropriate to address the stated objectives?

-Is the population clearly described and appropriate for the hypothesis being tested?

-Is the sample size sufficient to ensure adequate power to address the hypothesis being tested?

-Were correct statistical analysis used to support conclusions?

-Are there concerns about ethical or regulatory requirements being met?

Reviewer #1: Abstract: The abstract is well written, with key aspects of the study captured. Authors should clearly state the design, not the approach. Merely stating that you used qualitative research methods is not sufficient.

Introduction & background: The authors have made great efforts to produce a good background section. Some sentences are too lengthy, they could be revised or shortened.

Gaps and Objectives: Gaps have been clearly identified. The objectives need to come out clearly.

Methodology

Ethical considerations: Authors have ably written this section.

The study area has been addressed well highlighting key issues.

Research design: The design chosen is clear and aligns with the study objectives.

Sampling strategy: Authors have presented the sampling strategy in a concise and clear manner.

Data collection: This section is written well and brings out the methods and tools in a simple manner.

Data analysis: this section was properly written. However, authors could be more detailed on the analysis methods and procedures. As it is now, readers may not understand well how the thematic content analysis was done.

Reviewer #2: The methods applied are clear. However, the authors need to strengthen the inclusion criteria since it was purposive sampling. Note that they selected people with HIV and FGS; community health workers.

**Results**

-Does the analysis presented match the analysis plan?

-Are the results clearly and completely presented?

-Are the figures (Tables, Images) of sufficient quality for clarity?

Reviewer #1: Results: The results section is well presented with key findings highlighted. Some of the quotes do not depict the key findings. Eg on knowledge of FGS by KII R2 male, was not true. Instead, the KI participant quotes are about schistosomiasis in general, but not FGS.

Discussion: This section was well written and addresses all key findings.

Reviewer #2: The results are clearly presented. Authors can consider identifying key themes from the respondents to make the paper more concise.

**Conclusions**

-Are the conclusions supported by the data presented?

-Are the limitations of analysis clearly described?

-Do the authors discuss how these data can be helpful to advance our understanding of the topic under study?

-Is public health relevance addressed?

Reviewer #1: Conclusion and recommendations: The authors have made good conclusions and recommendations.

Reviewer #2: The conclusions are clear however the authors need to clearly state that there was no clinical evidence between FGS and HIV but instead that the study measured the knowledge, of FGS and HIV in a select group of people.

**Editorial and Data Presentation Modifications?**

Reviewer #1: Minor revision

Reviewer #2: Minor revision to improve clarity and ensure that the paper presents itself as measuring the knowledge of a sample of people in a country with high FGS prevalence. Statements that indicate a link between FGS and HIV should be clearly backed by evidence.

**Summary and General Comments**

Reviewer #1: General comments: This study makes significant contributions in the fight against both FGS and HIV/AIDs. Findings of this study could go a long way in improving management of FGS and HIV care not only in Ghana but Africa and the world.

Reviewer #2: The paper is a great contribution to a field of diseases (FGS) that are often not discussed or researched on by mainstream public health entities. It calls our attention to the need to better understand and advocate for the integration of FGS into HIV care and services. It also contributes to the global call to integrate FGS with sexual and reproductive health services.

PLOS authors have the option to publish the peer review history of their article (what does this mean? ). If published, this will include your full peer review and any attached files.

**Do you want your identity to be public for this peer review?** For information about this choice, including consent withdrawal, please see our Privacy Policy .

Reviewer #1: No

Reviewer #2: No

---

## [Editor Report · Decision Letter 1]

Dear Dr. Tabong,

We are pleased to inform you that your manuscript 'Stakeholders Perspective of Integrating Female Genital Schistosomiasis into HIV Care: A Qualitative Study in Ghana' has been provisionally accepted for publication in PLOS Neglected Tropical Diseases.

Best regards,

Hira L Nakhasi, Ph.D.

Section Editor

Hira Nakhasi

Section Editor

Shaden Kamhawi

co-Editor-in-Chief

Paul Brindley

co-Editor-in-Chief

The authors have satisfactorily revised the manuscript based on the reviewer's comments.

---

## [Editor Report · Acceptance letter]

Dear Dr. Tabong,

We are delighted to inform you that your manuscript, "Stakeholders Perspective of Integrating Female Genital Schistosomiasis into HIV Care: A Qualitative Study in Ghana," has been formally accepted for publication in PLOS Neglected Tropical Diseases.

Best regards,

Shaden Kamhawi

co-Editor-in-Chief

Paul Brindley

co-Editor-in-Chief
